# Assessing the burden of severe nausea and vomiting of pregnancy or hyperemesis gravidarum and the associated use and experiences of medication treatments: An Australian consumer survey

Loyola Wills[1], Han-Fang Hsiao[2], Alicia Thomas[2], Caitlin Kay-Smith[3], Amanda Henry[4,5], Luke E. Grzeskowiak[1,2,6,7] *

1 College of Medicine and Public Health, Flinders University, Adelaide, South Australia, Australia, 2 SA Pharmacy, Flinders Medical Centre, SA Health, Adelaide, Australia, 3 Hyperemesis Australia, Sydney, New South Wales, Australia, 4 University of New South Wales, Sydney, New South Wales, Australia, 5 St George Hospital, Sydney, New South Wales, Australia, 6 Adelaide Medical School, Robinson Research Institute, University of Adelaide, Adelaide, Australia, 7 SAHMRI Women and Kids, South Australian Health and Medical Research Institute, Adelaide, Australia

* luke.grzeskowiak@flinders.edu.au

## Abstract

### Background

There is little data on contemporary patterns of antiemetic use or women's experiences when using such agents in the treatment of severe nausea and vomiting of pregnancy (NVP) or hyperemesis gravidarum (HG).

### Methods

Online, national survey of Australian women who were currently or had previously experienced severe NVP or HG, distributed through the HG consumer group, Hyperemesis Australia between July and September 2020.

### Results

There were a total of 289 respondents with a mean age of 33 years, of which 38% were currently pregnant. More than 50% of respondents reported "major impacts" of the condition on areas such as social life, ability to undertake daily chores, ability to eat or drink, effects on work, taking care of pre-existing children and sleep. This resulted in 62% of respondents reporting 'often' or 'always' experiencing feelings of depression or anxiety as a result of their HG symptoms, with 54% reporting considering terminating their pregnancy, and 90% having considered having no more children. The most commonly used anti-emetic was ondansetron (91%), followed by pyridoxine (62%), doxylamine (62%), and metoclopramide (61%). Nearly all (95%) women

**Data availability statement:** Data cannot be shared publicly because of privacy restrictions imposed by the University of Adelaide Human Research Ethics Committee. Requests for data must be made to the University of Adelaide Human Research Ethics Committee (hrec@adelaide.edu.au) who may grant requests for researchers who meet the criteria for access to confidential data.

**Funding:** This study was supported by a 2020 Engaging Opportunities research grant provided by the Robinson Research Institute, The University of Adelaide. LEG receives salary support from the Channel 7 Children's Research Foundation (CRF-210323). The funding source(s) had no involvement in the conduct of the research and/or preparation of the article.

**Competing interests:** The authors declare that they have no competing interests.

who reported using ondansetron commenced it within the first trimester, with 55% reporting use as a first-line therapy. Most women reported one or more side effects to anti-emetics such as headache, constipation, sedation or impaired cognition, with 31% stopping metoclopramide because of side effects, compared with 14% for ondansetron and 10% for doxylamine. Ondansetron, doxylamine and corticosteroids had the greatest perceived effectiveness, with more than 50% rating them as "effective" or "very effective". Half (50%) reported use of acid suppressive therapy, with 51% reporting using complementary or alternative therapies in addition to conventional treatments.

## Conclusions

The study findings demonstrate large variability in antiemetic use and outcomes, highlighting the need for individualised care and treatment approaches during pregnancy.

## Introduction

Many women report experiencing Nausea and/or Vomiting of pregnancy (NVP) with varying degrees of severity. Hyperemesis Gravidarum (HG) is a severe form of NVP, that can lead to hypovolemia, electrolyte imbalances and weight loss leading to severe maternal-fetal outcomes [1]. HG is reported to affect 0.3–10.8% of pregnancies [2]. While previously not well defined, a recently developed International consensus definition for HG, the Windsor definition, requires the following to be met: symptoms commencing prior to 16 weeks' gestation, severe nausea and/or vomiting, inability to eat and/or drink normally and strongly limiting daily activities [3]. Signs of dehydration or weight loss were not considered mandatory to diagnosing HG. The etiology of HG is unknown, but is considered multifactorial in nature, with emerging data regarding genetic predispositions [2,4]. HG has been associated with an increased risk of adverse maternal, fetal and child outcomes [2]. Further, studies have previously shown that HG significantly affects maternal wellbeing, including social, emotional and occupational aspects [5]. Therefore, the early identification and effective treatment of HG with medications is critical to reducing associated negative sequalae, which is reflected across a number of local and international guidelines and practice standards [1,6–8]. Most guidelines recommend using the PUQE-24 (Pregnancy Unique Qualification of Emesis Scale) scoring system to assist in identification and step-wise management of NVP/HG. PUQE-24 characterises NVP/HG as mild, moderate and severe, based on three criteria: hours of nausea, number of episodes of retching and number of episodes of vomiting within the last 24 hours [9]. Alternative, albeit lengthier, assessment tools have since been published and shown to be as effective or better at differentiating more severe forms of NVP/HG [10].

While treatments for HG are reflected in a step-wise manner in guidelines, treatment should be directed at the severity of symptoms at the time of presentation,

which may necessitate commencing treatments at a higher step rather than working through in a step-wise manner [1,6–8]. Given there is limited high-quality evidence to support one anti-emetic over another, management of HG often involves trialing or utilizing multiple anti-emetic agents to determine which are effective and able to be tolerated. This commonly involves the use of agents such as doxylamine, metoclopramide, and ondansetron. In situations where symptoms are refractory to existing treatment, corticosteroids may be used. Most women with HG can be adequately managed in the outpatient setting, but some may require admission to hospital, particularly where intravenous fluid or electrolyte replacement is required, or enteral or parenteral nutrition approaches are needed. Consideration also needs to be provided towards the use of supportive therapies such as laxatives or acid suppression, and the use of vitamin supplements to support nutritional requirements.

Despite the presence of guidelines supporting adequate assessment and treatment of HG, previous studies have shown that some health care professionals still demonstrate reluctance to prescribe anti-emetics during pregnancy, [11] commonly downplaying symptoms women experience as being a 'normal part of pregnancy [12]. This necessitates frequent and ongoing evaluations of how HG is being assessed and managed in clinical practice, however, no recent studies have explored women's experiences with respect to treatments for HG and how this compares to current practice guidelines. Therefore, the aim of this study was to conduct a survey of women affected by HG regarding what treatments they used, how they used them, and what their experience with treatments were in terms of perceived effectiveness and safety.

## Materials and methods

### Study design and data collection

We carried out a cross-sectional online survey study. Women currently living in Australia and who were currently or had previously experienced severe NVP or HG were eligible to complete the survey. Women could be pregnant at the time of completing the survey. The survey was available online between July and September 2020. The survey consisted of three parts: part A, respondents characteristics; part B, awareness and perceived safety of treatments for HG as well as use of information sources and their perceived usefulness; Part C, characteristics of HG, symptoms, and impacts on quality of life; and Part D, personal experiences and use of treatments for HG, including perceived effectiveness, side effects and timing and duration of use. This paper focusses on Parts A, C and D, with Part B covered as part of a separate analysis. Survey data included:

- Respondent demographics, including age, racial or ethnic group, marital status, education, gravida and parity, smoking status, and chronic illnesses.

- Hyperemesis medical history, including number of affected pregnancies, timing of onset of symptoms and when symptoms ceased, PUQE24 score when symptoms were at their worst during pregnancy, weight loss, difficulty eating and/or drinking, hospital presentations and admission.

- Impacts of HG on quality of life, including Likert scale questions from 1 labelled as "not at all" and 5 labelled as "major impact" related to "Inhibition of the ability to take care of household chores", "reduced social life", "negative impact on relationship with partner", "reduced work capacity", "the ability to care for any children from previous pregnancies", "having uninterrupted sleep", and "inhibition of the ability to eat or drink". Further questions were also explored including consideration of termination of pregnancy due to NVP, experiences of depression and/or anxiety stemming from the NVP/HG, if these feelings led to request for induction or elective cesarean section and the consideration of not having any more children due to NVP/HG.

- Use and experiences of treatments for HG, including what the treatment was, what week of pregnancy it was started and/or stopped, duration of treatment, whether side effects were experienced, if treatment was stopped due to side effects, and the perceived effectiveness of treatment on a Likert scale from 1 (not very effective) to 5 (very effective).

The survey was tested for face validity with two consumer representatives and two clinical experts. Only minor changes were made to the survey before formal distribution through the social networks (i.e., website, Facebook, Twitter) of the consumer organization, Hyperemesis Australia, as well as research networks of the author's respective institutions (e.g., The Robinson Research Institute, and The University of Adelaide). Participants were encouraged to share the survey and post links to the survey through their own social networks. The complete survey is available as S1 Appendix.

Study data were collected and managed using Research Electronic Data Capture (REDCap) hosted at The University of Adelaide [13,14]. REDCap is a secure, web-based software platform designed to support data capture for research studies, providing 1) an intuitive interface for validated data capture; 2) audit trails for tracking data manipulation and export procedures; 3) automated export procedures for seamless data downloads to standard statistical packages, and 4) procedures for data integration and interoperability with external sources. Only study investigators involved in the study had access to the data.

Completing the survey was voluntary, and no incentives were offered to participants. Individual IP addresses of respondents were not tracked.

## Statistical analyses

Data were cleaned and analysed using Stata 16 (StataCorp LP, College Station, TX). Graphical images were produced using GraphPad Prism version 9 (GraphPad Software, La Jolla, California, USA) and the R UpSet package [15]. We first searched for duplicate entries based on identical maternal characteristics provided in the entry section in order to remove any potential duplicate responses. Participants may have chosen not to answer every question in the survey, with the analysis undertaken based on available data. No weighting of the data was applied.

The sociodemographic characteristics of women responding to the survey and data on medication use and experiences of treatment were described using descriptive statistics. Experiences of medication treatment are only presented for medications that were reported to be used by 10 or more people. The most common combinations of anti-emetic agents used by individuals throughout pregnancy were graphed using an UpSet plot. Given the non-normal distribution of timing of treatment initiation and duration of use, these were reported using medians and interquartile ranges and presented graphically using a box-plot. Differences between groups were examined using chi-square statistic, with $p < 0.05$ used to define statistical significance.

## Ethics

The study was approved by the University of Adelaide's Human Research Ethics Committee (H-2020-090) and conducted according to the principles expressed in the Declaration of Helsinki. Participants provided fully informed implied consent before participating anonymously. Contact details were only required if participants chose to complete a follow-up interview as part of a separate qualitative research study.

## Results

### Maternal demographics and clinical characteristics

There were a total of 289 respondents to the survey. The mean age of respondents at the time of completing the survey was 33 (SD 5.8 years), of which 110 (38%) were currently pregnant. Most respondents identified as Caucasian (265, 94%), were married or in a de facto relationship (272, 95%) and had completed secondary schooling (246, 87%). The majority had been pregnant two or more times (231, 80%) and had given birth one or more times (254, 88%). Most respondents did not smoke during their current/previous pregnancy (277, 96%).

With respect to their most recent pregnancy affected by HG, 110 (38%) indicated they were currently pregnant and experiencing HG, 47 (16%) that their pregnancy ended <6 months ago, 43 (15%) that it ended 6–12 months ago, 39

(14%) that it ended 1–2 years ago and 50 (17%) that it ended >2 years ago. The median onset of NVP was 6 weeks' gestation, with everyone reporting onset within the first trimester. Most women (218, 75%) reported experiencing weight loss during pregnancy, with a further 29 (10%) being unsure. Among those who reported losing weight, the median weight loss was 7 kg, ranging from 1 to 40 kg. The PUQE-24 score classified 118 (41%) and 169 (59%) as experiencing moderate or severe NVP respectively. Most respondents (224, 76%) reported receiving a formal diagnosis of HG at some stage during pregnancy. Two-hundred and seven (72%) respondents reported being admitted to hospital for IV fluids at some stage during pregnancy, while nearly all respondents (283, 98%) reported experiencing great difficult eating and drinking as normal during pregnancy. One-hundred and seventy-nine (62%) respondents reported 'often' or 'always' experiencing feelings of depression or anxiety as a result of their HG symptoms. One-hundred and eight (37%) respondents reported requesting an induction of labour to bring forward delivery as a result of their HG symptoms, while a total of 155 (54%) respondents reported considering terminating their pregnancy, and 259 (90%) have considered having no more children.

The self-reported impacts of HG on everyday aspects of life are reported in Fig 1. More than 50% of respondents reported major impacts on areas such as social life, ability to undertake daily chores, ability to eat or drink, effects on work, taking care of pre-existing children and sleep. While smaller in magnitude, 30% also reported negative impacts on their relationship with their partner.

## Treatment used

All women reported using at least one antiemetic during pregnancy, with the mean number of anti-emetics taken being 4.2, ranging from 1 to 9. Overall frequencies and combinations of different anti-emetics are presented in Fig 2. The most common anti-emetic was ondansetron (263, 91%), followed by pyridoxine (202, 70%), doxylamine (201, 70%), metoclopramide (200, 69%), and ginger (152, 53%). The most common combination used (n = 33) was ginger, metoclopramide, doxylamine, pyridoxine and ondansetron. Followed by doxylamine, pyridoxine and ondansetron (n = 16), which is equal with Metoclopramide, doxylamine, pyridoxine and ondansetron. Only 20 (7%) of respondents reporting using a single anti-emetic during pregnancy.

Data on the median timing of treatment initiation and total duration of use during pregnancy for each anti-emetic agent are reported in Fig 3. While a large range was evident in terms of timing of treatment initiation and duration of use, medications with the earliest treatment initiation were ginger, metoclopramide, ondansetron and pyridoxine with median times each of 6 weeks' gestation. The median time of initiation of corticosteroids was latest, at 12 weeks' gestation. Ginger, metoclopramide, and prochlorperazine were used for the shortest duration (median of 2–4 weeks' gestation). Doxylamine and ondansetron had the longest reported durations of use, with medians of 16 and 20 weeks' respectively.

The proportion of respondents indicating whether the anti-emetic was used first-line either as the sole agent, or in combination with another treatment, or as a second or subsequent treatment is reported in Fig 4. Ginger was the most common medication initiated as first-line treatment, either on its own, or as part of combination therapy in 88%, followed

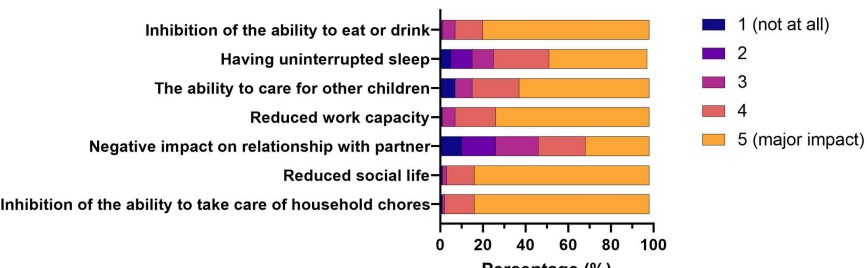

**Fig 1. Negative impacts of experiencing severe nausea and vomiting or hyperemesis on activities of everyday life.**

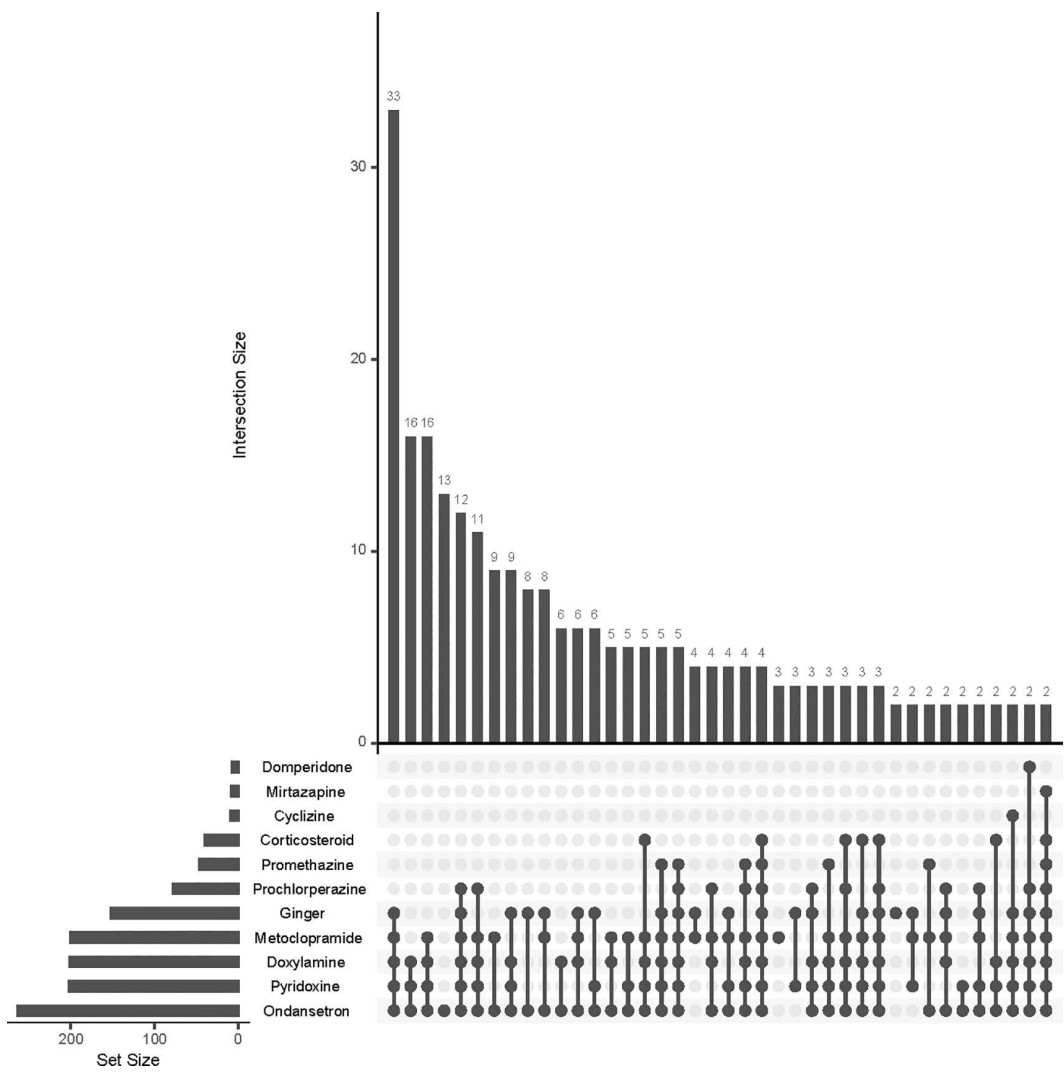

**Fig 2. UpSet plot showing the use of different anti-emetic agents and combinations.**

by ondansetron (55%), metoclopramide (49%) and pyridoxine (46%). Notably, first-line ondansetron use was more likely to be reported by those experiencing their second or subsequent pregnancy affected by HG, compared with those experiencing their first affected pregnancy (64% vs. 27%, p = 0.015).

## Side effects

Side effects women experienced according to anti-emetic treatment are presented in **Table 1**. Doxylamine (78%), ondansetron (73%) and promethazine (72%) each had the highest proportion of women reporting one or more side effects. Medications most likely to be ceased as a result of side effects included metoclopramide (31%), ginger (24%), and prochlorperazine (23%). Among the more commonly used medications, the most common side effect with ondansetron was constipation (65%), compared with sedation for doxylamine (65%), mood disorders with metoclopramide (16%), and constipation with pyridoxine (12%).

**A**

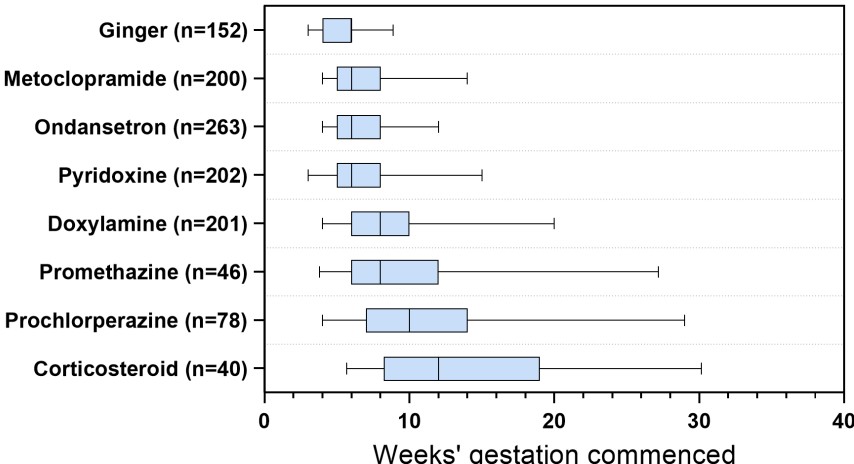

**B**

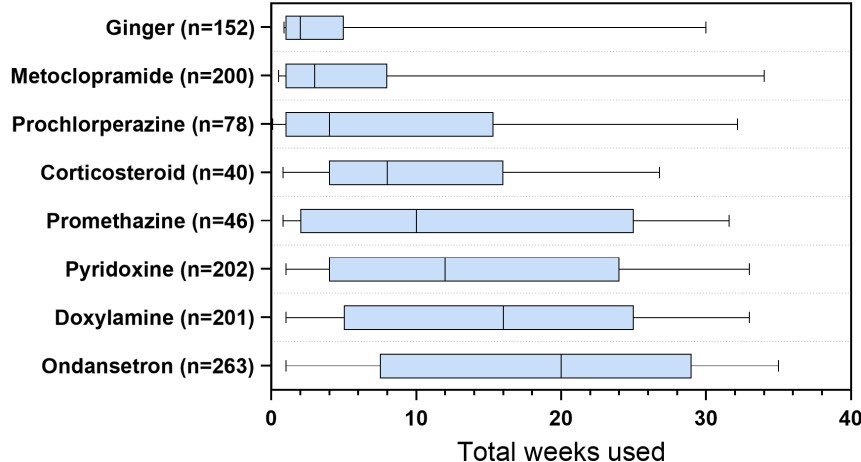

**Fig 3.** Median and interquartile range for (a) timing of treatment initiation, (b) total duration of treatment.

More than half the participants reported issues with constipation for ondansetron, sedation for doxylamine and less so, promethazine. The least reported side effect was mood disorder, however this was still seen in some participants taking metoclopramide, prochlorperazine and corticosteroids.

### Perceived effectiveness

The perceived effectiveness of anti-emetic medications is reported in **Fig 5**. Medications rated as being the most effective were corticosteroids, ondansetron, and doxylamine with more than 50% of respondents rating them as being "effective" or "very effective". In contrast, less than 10% of respondents reported pyridoxine or ginger as being "effective" or "very effective".

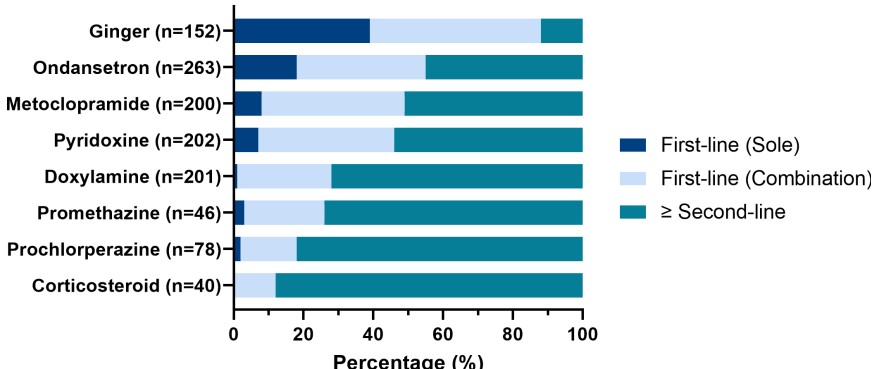

**Fig 4. Percentage of respondents commencing anti-emetics as first- or second-line therapy.**

**Table 1. Self-reported side effects for anti-emetic medications.**

|  | Corticosteroid | Promethazine | Prochlorperazine | Ginger | Metoclopramide | Doxylamine | Pyridoxine | Ondansetron |
|---|---|---|---|---|---|---|---|---|
| **Took medication (N)** | 40 | 46 | 78 | 152 | 200 | 201 | 202 | 263 |
| **Any side effect[a]** | 24 (60) | 33 (72) | 35 (45) | 57 (38) | 102 (51) | 156 (78) | 60 (30) | 191 (73) |
| **Ceased due to side effect[a]** | 2 (5) | 9 (20) | 18 (23) | 36 (24) | 61 (31) | 20 (10) | 18 (9) | 36 (14) |
| **Individual Symptoms[a]** |  |  |  |  |  |  |  |  |
| Headache | 10 (25) | 4 (9) | 9 (12) | 8 (5) | 25 (13) | 24 (12) | 12 (6) | 42 (16) |
| Muscle/body weakness | 6 (15) | 3 (7) | 4 (5) | 3 (2) | 14 (7) | 12 (6) | 5 (2) | 15 (6) |
| Numbness | 2 (5) | 1 (2) | 3 (4) | 2 (1) | 6 (3) | 4 (2) | 3 (1) | 7 (3) |
| Stomach Cramps | 1 (3) | 0 | 1 (1) | 6 (4) | 8 (4) | 2 (1) | 8 (4) | 29 (11) |
| Heartburn | 2 (5) | 2 (4) | 1 (1) | 29 (19) | 8 (4) | 9 (4) | 13 (6) | 19 (7) |
| Constipation | 3 (8) | 2 (4) | 3 (4) | 8 (5) | 23 (12) | 24 (12) | 24 (12) | 172 (65) |
| Diarrhoea | 1 (3) | 0 | 1 (1) | 0 | 4 (2) | 2 (1) | 4 (2) | 6 (2) |
| Impaired Cognition | 4 (10) | 10 (22) | 8 (10) | 2 (1) | 20 (10) | 40 (20) | 3 (1) | 16 (6) |
| Sedation | 2 (5) | 26 (57) | 12 (15) | 1 (1) | 24 (12) | 130 (65) | 4 (2) | 23 (9) |
| Worsening N&V | 2 (5) | 4 (9) | 6 (8) | 36 (24) | 26 (13) | 4 (2) | 20 (10) | 13 (5) |
| Dry Mouth | 6 (15) | 3 (7) | 6 (8) | 6 (4) | 19 (10) | 33 (16) | 14 (7) | 33 (13) |
| Blurred Vision | 1 (3) | 1 (2) | 1 (1) | 0 | 5 (3) | 2 (1) | 1 (0) | 2 (1) |
| Other | 3 (8) | 0 | 13 (17) | 0 | 4 (2) | 2 (1) | 0 | 5 (2) |
| Mood Disorder | 5 (13) | 0 | 11 (14) | 0 | 32 (16) | 3 (1) | 1 (0) | 0 |

[a]n (% of those who took each medication)

## Discussion

In this relatively large survey of Australian women experiencing severe NVP or HG, we demonstrate widespread variation in use and effects of anti-emetic treatments, with evidence of increasing first-line treatment with ondansetron, particularly within the first trimester of pregnancy. The survey also confirms previously reported widespread negative impacts of HG on activities of daily life, further highlighting the importance of appropriate recognition and treatment of symptoms with safe and effective therapies.

The identified rates of anti-emetic medication use are higher than reported in previous survey studies [12,16]. In their online survey of 712 Norwegian women back in 2014–2015, Heitmann et al. identified that 67.1% of respondents with

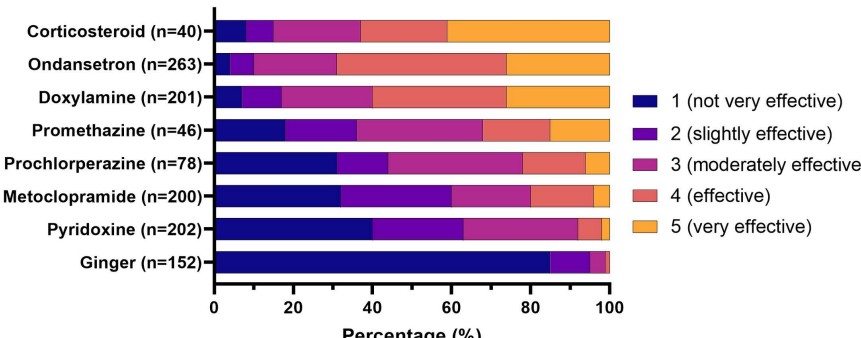

**Fig 5. Perceived effectiveness of anti-emetic medications.**

severe NVP reported using one or more medications [12], compared with 100% in our survey. Further, in their 2020 online survey of women with HG (87% of respondents from the United States of America) Mares et al. identified that 57% reported taking three or more anti-emetics [16], compared with 81% in our study. Mares et al also identified ondansetron as the most commonly used anti-emetic, use was reported by approximately 57%, compared with 91% in our study (and <10% by Heitmann et al). Observed differences with previous studies could be reflective of differences in patient demographics or illness severity, regional differences in assessment and management of HG, and/or changes in practice over time.

While the findings of our survey are largely in accordance with current Australian clinical practice guidelines [6], the results do shed an important light on what is actually occurring in clinical practice. Due to conflicting data as to whether or not ondansetron increases the risk for certain congenital malformations, it is typically recommended for use as a second-line therapy [17,18]. Notably, 50% of respondents reporting using ondansetron as a first-line therapy. Similarly to ondansetron, guidelines commonly recommend avoiding the use of corticosteroids in the first trimester, where possible, due to a potential increased risk of oral clefts [19], but again evidence is conflicting. In our survey, approximately 50% of corticosteroid users reported commencing it in the first trimester. Given the conflicting data that exists as to whether or not ondansetron or corticosteroids do truly increase the risk for malformations or not, the frequent and apparently increasing use of these agents in the first trimester highlights the need for better evidence to guide future practice. Metoclopramide represents another anti-emetic where potential safety concerns exist, but in this case it relates to potential maternal harm. Regulatory recommendations limit duration of treatment with metoclopramide to less than five days due to potential increased risk for extrapyramidal side effects [8]. While the median reported treatment duration observed in our study was relatively short (3 weeks), this appears to be related to a significant number of women experiencing treatment limiting side effects including mood disorders. This can be compared with ondansetron, which while having a much higher proportion reporting side effects, was less than half as likely to be ceased as a result of side effects than metoclopramide. Taken together, the potential for harm from anti-emetics must be weighed against the harm of uncontrolled symptoms and individuals supported to make informed decisions about treatments [8].

A recent systematic review concluded that routinely prescribed anti-emetics including antihistamines, metoclopramide, pyridoxine-doxylamine, and ondansetron had superior efficacy to placebo [20]. However, data comparing the efficacy of different anti-emetics remains limited [21]. With emerging evidence that ondansetron is superior to both doxylamine/pyridoxine in reducing symptoms of nausea and vomiting [22], as well as superior to metoclopramide in reducing symptoms of vomiting, or is at least similarly effective with less side effects [23,24], it is perhaps unsurprising that it is increasingly being used as a first-line therapy in clinical practice. This evidence translated to strong perceptions of ondansetron being highly effective in those prescribed it, while appearing to be relatively well tolerated. This contrasts to the majority of other

anti-emetics that can have significant negative impacts on mood, alertness (i.e., sedation) and cognition, which do appear to significantly limit their overall use.

Clinical guidelines also highlight the importance of supportive therapies such as acid suppression for the management of severe NVP and HG [6]. There is evidence that treatment of gastro-oesophageal reflux along with anti-emetic therapy is associated with reduced PUQE-24 scores (9.6±3.0 to 6.5±2.5, P<.0001) and improved quality of life scores (4.0±2.0 to 6.8±1.6, p<.0001) [25]. Despite this, we identified that only 30% of women reported use of H2 antagonists or PPIs, albeit much higher than reported by Mares et al (<5%) [16]. Therefore, ensuring clinicians are aware of the importance of acid suppressive therapy and recommending their use appropriately warrants further attention.

The observed significant negative impacts of HG on activities of daily living and mental health are consistent with that reported in previous quantitative [12,26] and qualitative [27–29] studies. The significance of such impacts are reflected in 54% of our respondents reporting considering terminating their pregnancy as a result of HG, which is very similar to the rate of 52% reported by Nana et al in their recently published survey of 5071 women from the United Kingdom [30]. While we didn't collect data on whether any pregnancies were terminated as a result of HG, 37% reported requesting an early induction of labour to bring forward delivery. We are not aware of this being evaluated in previous surveys. Previous studies have shown that whilst quality of life can be enhanced by the use of medications [31], supportive care and psychological input is necessary given that some women can find the symptoms of NVP distressing enough where suicidal ideation has been reported [32].

## Strengths and limitations

Strengths of this survey include the focus on a cohort of women with severe NVP or HG, as well as the comprehensive range of data collected regarding treatments and experiences. While this study provides valuable insights it is not without limitations. Women self-identified as having experienced severe NVP or HG during pregnancy, albeit during the time of the study no agreed international definition for severe NVP or HG existed. Our survey used non-probabilistic sampling and snowballing sampling techniques, making it difficult to extrapolate findings to the broader Australian pregnant population. Those who chose to respond to the survey may differ in some way from those that chose not to respond. The survey measures women's perceived effectiveness of treatments and did not utilise objective measures of changes in symptoms. Respondents included a mix of women who were currently pregnant and those reflecting on previous pregnancies, which in some cases ended more than 2-years prior. This raises the possibility of errors related to some women's ability to accurately recall exact timings and duration of use, or each individual anti-emetic therapy that they used. Furthermore, for women who were still pregnant at the time of completing the survey, it is not possible to correctly define the total duration of medication use or their complete set of experiences related to medication use.

## Conclusions

Our study sheds light on the complex landscape of medication use for severe NVP and HG, demonstrating significant clinical variation in individual use and experiences of treatments. Our findings also emphasise the profound psychosocial burden experienced by women with severe NVP and HG, and the significant impacts living with this condition can also have on quality of life. Our research underscores the need for a nuanced and holistic approach to clinical management of severe NVP and HG. While guidelines provide a structured framework to guide clinical management, care should be individualised based on a thorough understanding of patient preferences, with due recognition for the associated psychosocial impacts of the illness and its treatment. Greater evidence is also urgently needed to help better guide existing and future treatments for this debilitating condition, particularly with respect to exploring immediate and long-term impacts of various medications on maternal, fetal, and infant health and wellbeing.

## Supporting information

**S1 Appendix. Hyperemesis Survey.**
(PDF)

## Author contributions

**Conceptualization:** Alicia Thomas, Caitlin Kay-Smith, Luke E. Grzeskowiak.

**Data curation:** Han-Fang Hsiao, Luke E. Grzeskowiak.

**Formal analysis:** Loyola Wills, Luke E. Grzeskowiak.

**Funding acquisition:** Caitlin Kay-Smith, Luke E. Grzeskowiak.

**Investigation:** Loyola Wills, Han-Fang Hsiao, Luke E. Grzeskowiak.

**Methodology:** Han-Fang Hsiao, Alicia Thomas, Caitlin Kay-Smith, Amanda Henry, Luke E. Grzeskowiak.

**Supervision:** Luke E. Grzeskowiak.

**Writing – original draft:** Loyola Wills.

**Writing – review & editing:** Han-Fang Hsiao, Alicia Thomas, Caitlin Kay-Smith, Amanda Henry, Luke E. Grzeskowiak.

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
