## [Decision Letter · Decision Letter 0]

PONE-D-25-16415Assessing the burden of severe nausea and vomiting of pregnancy or hyperemesis gravidarum and the associated use and experiences of medication treatments: an Australian consumer surveyPLOS ONE

Dear Dr. Grzeskowiak,

Thank you for submitting your manuscript to PLOS ONE. After careful consideration, we feel that it has merit but does not fully meet PLOS ONE’s publication criteria as it currently stands. Therefore, we invite you to submit a revised version of the manuscript that addresses the points raised during the review process.

We look forward to receiving your revised manuscript.

Kind regards,

Shipra Sonkusare, MD, DNB, MNAMS, FICOG, MRCOG, FRCOG

Academic Editor

PLOS ONE

Journal Requirements:

“This study was supported by a 2020 Engaging Opportunities research grant provided by the Robinson Research Institute, The University of Adelaide. LEG receives salary support from the Channel 7 Children’s Research Foundation (CRF-210323). The funding source(s) had no involvement in the conduct of the research and/or preparation of the article.”

“This study was supported by a 2020 Engaging Opportunities research grant provided by the Robinson Research Institute, The University of Adelaide. LEG receives salary support from the Channel 7 Children’s Research Foundation (CRF-210323). The funding source(s) had no involvement in the conduct of the research and/or preparation of the article.”

“This study was supported by a 2020 Engaging Opportunities research grant provided by the Robinson Research Institute, The University of Adelaide. LEG receives salary support from the Channel 7 Children’s Research Foundation (CRF-210323). The funding source(s) had no involvement in the conduct of the research and/or preparation of the article.”

7. Please remove your figures from within your manuscript file, leaving only the individual TIFF/EPS image files, uploaded separately. These will be automatically included in the reviewers’ PDF.

Reviewers' comments:

Reviewer's Responses to Questions

**Comments to the Author**

1. Is the manuscript technically sound, and do the data support the conclusions?

Reviewer #1: Yes

Reviewer #2: Partly

2. Has the statistical analysis been performed appropriately and rigorously? 

Reviewer #1: Yes

Reviewer #2: Yes

3. Have the authors made all data underlying the findings in their manuscript fully available?

Reviewer #1: No

Reviewer #2: Yes

4. Is the manuscript presented in an intelligible fashion and written in standard English?

Reviewer #1: Yes

Reviewer #2: Yes

5. Review Comments to the Author

Reviewer #1: This is an insightful study that provides valuable new perspectives on the management of nausea and vomiting during pregnancy.

The abstract effectively summarizes the main concepts and objectives of the study.

The introduction provides a comprehensive background and clearly outlines the objective of the research.

However, the methodology section raises a few points that require clarification:

1. Could the authors specify how the responders were selected for the study?

2. Please provide further details regarding the validity and reliability of the questionnaire used.

3. When were the responders approached to complete the questionnaire? Was it during their pregnancy? If so, at which trimester?

The results are presented in a clear and organized manner.

The discussion is thorough and adequately addresses the key findings of the study.

Reviewer #2: The authors reported results about an online, national survey of 289 Australian women who were currently or had previously experienced severe NVP or HG, distributed through the HG consumer group, Hyperemesis Australia between July and September 2020. They showed a large variability in antiemetic use and outcomes, highlighting the need for individualized care and treatment approaches during pregnancy.

The subject is interesting because very few published data reported results about national survey in women with NVP or HG. Nevertheless, the study contains many flaws/bias that decrease the interest for the study and major modifications should be done for more clarity.

First, a BIG bias: the survey was realized in women with severe NVP or HG through the HG consumer group, Hyperemesis Australia. Some affirmations were quite evident: “More than 50% of respondents reported major impacts...”, and some affirmations should be written with caution and more discussed: “54% reporting considering terminating their pregnancy and 90% having considered having no more children”. Nevertheless, only 59% of included women were classified as severe NVP using the PUQE score. So, who are the included women????? Are the conclusions adapted???

Second, the recruitment method induced another bias in results such as treatments used in fist line (91% ondansetron and 95% in first trimester) because only severe NVP or HG were included. Many of included women had also experienced HG in a previous pregnancy with ondansetron (64%) as a second or a third line treatment in this previous pregnancy and was then directly used ondansetron as first line in a new pregnancy with HG. It is quite logical!

The authors should clarify who were the included women: severe NPV or HG, only HG, all women with NVP..). In abstract, in the conclusion, “in cases of severe NVP or HG” should be added to recontextualize. In methods, in “study design”, it is written: “Women currently living in Australia and who were currently or had previously experienced HG were eligible to complete the survey”. Severe NVP as an inclusion criterion has been deleted!

The discussion is too long, should be shortened, and bias should be more discussed.

6. PLOS authors have the option to publish the peer review history of their article (what does this mean? ). If published, this will include your full peer review and any attached files.

**Do you want your identity to be public for this peer review?** For information about this choice, including consent withdrawal, please see our Privacy Policy .

Reviewer #1: No

Reviewer #2: No

---

## [Author Response · Author response to Decision Letter 1]

9 Jul 2025

Editor Comments

Comment 1. Please ensure that your manuscript meets PLOS ONE's style requirements, including those for file naming. The PLOS ONE style templates can be found at

Response 1. We have carefully reviewed to ensure the manuscript has been formatted in accordance with journal guidelines (formatting changes not tracked)

Comment 2. Thank you for stating in your Funding Statement:

“This study was supported by a 2020 Engaging Opportunities research grant provided by the Robinson Research Institute, The University of Adelaide. LEG receives salary support from the Channel 7 Children’s Research Foundation (CRF-210323). The funding source(s) had no involvement in the conduct of the research and/or preparation of the article.”

Response 2. Updated funding statement: “This study was supported by a 2020 Engaging Opportunities research grant provided by the Robinson Research Institute, The University of Adelaide. LEG receives salary support from the Channel 7 Children’s Research Foundation (CRF-210323). The funding source(s) had no involvement in the conduct of the research and/or preparation of the article. There was no additional external funding received for this study.”

Comment 3. We note that you have indicated that there are restrictions to data sharing for this study. For studies involving human research participant data or other sensitive data, we encourage authors to share de-identified or anonymized data. However, when data cannot be publicly shared for ethical reasons, we allow authors to make their data sets available upon request. For information on unacceptable data access restrictions, please see http://journals.plos.org/plosone/s/data-availability#loc-unacceptable-data-access-restrictions.

Response 3. We have updated the data sharing statement:

“Data cannot be shared publicly because of privacy restrictions imposed by the University of Adelaide Human Research Ethics Committee. Requests for data must be made to the University of Adelaide Human Research Ethics Committee (hrec@adelaide.edu.au) who may grant requests for researchers who meet the criteria for access to confidential data.”

Comment 4. Thank you for stating the following in the Acknowledgments Section of your manuscript:

“This study was supported by a 2020 Engaging Opportunities research grant provided by the Robinson Research Institute, The University of Adelaide. LEG receives salary support from the Channel 7 Children’s Research Foundation (CRF-210323). The funding source(s) had no involvement in the conduct of the research and/or preparation of the article.”

“This study was supported by a 2020 Engaging Opportunities research grant provided by the Robinson Research Institute, The University of Adelaide. LEG receives salary support from the Channel 7 Children’s Research Foundation (CRF-210323). The funding source(s) had no involvement in the conduct of the research and/or preparation of the article.”

Response 4. We apologise for this oversight. The acknowledgement section should state: “None”, with all relevant information contained within the funding statement.

Comment 5. We note that you have included the phrase “data not shown” in your manuscript. Unfortunately, this does not meet our data sharing requirements. PLOS does not permit references to inaccessible data. We require that authors provide all relevant data within the paper, Supporting Information files, or in an acceptable, public repository. Please add a citation to support this phrase or upload the data that corresponds with these findings to a stable repository (such as Figshare or Dryad) and provide and URLs, DOIs, or accession numbers that may be used to access these data. Or, if the data are not a core part of the research being presented in your study, we ask that you remove the phrase that refers to these data.

Response 5. We apologise for this oversight. We have removed the phrase that refers to these data.

Comment 6. Your ethics statement should only appear in the Methods section of your manuscript. If your ethics statement is written in any section besides the Methods, please delete it from any other section.

Response 6. The ethics statement at the end of the manuscript has been removed and now only contained within the methods section.

Comment 7. Please remove your figures from within your manuscript file, leaving only the individual TIFF/EPS image files, uploaded separately. These will be automatically included in the reviewers’ PDF.

Response 7. Figures have been removed from the manuscript file.

Reviewer 1

Comment 8. This is an insightful study that provides valuable new perspectives on the management of nausea and vomiting during pregnancy.

The abstract effectively summarizes the main concepts and objectives of the study.

The introduction provides a comprehensive background and clearly outlines the objective of the research.

Response 8. Thank you, no amendments necessary

Comment 9. However, the methodology section raises a few points that require clarification:

Could the authors specify how the responders were selected for the study?

Response 9. As described in the methods, “the survey was distribution through the social networks (i.e. website, Facebook, Twitter) of the consumer organization, Hyperemesis Australia, as well as research networks of the author’s respective institutions (e.g. The Robinson Research Institute, and The University of Adelaide). Participants were encouraged to share the survey and post links to the survey through their own social networks.”

We have also acknowledged this as a study limitation:

“Our survey used non-probabilistic sampling and snowballing sampling techniques, making it difficult to extrapolate findings to the broader Australian pregnant population. Those who chose to respond to the survey may differ in some way from those that chose not to respond.”

Comment 10. Please provide further details regarding the validity and reliability of the questionnaire used. Aside from individual questionnaires like the PUQE score, there are no questionnaires validated for assessing individual’s experiences and perspectives on medications for managing severe NVP or HG.

Response 10. We largely based our questions on those included in the online survey by Heitmann et al (Heitmann K, Solheimsnes A, Havnen GC, Nordeng G, Holst L. Treatment of Nausea and vomiting during pregnancy – a cross sectional study among 712 Norwegian women. Eur J Clin Pharmacol. 2016;72:593-604.). This was to provide the ability to compare our results to previous studies.

The survey was also co-designed with our consumer representative organization (Hyperemesis Australia).

We have included a description in the methods regarding the survey: “The survey was tested for face validity with two consumer representatives and two clinical experts.”

A copy of the complete survey is also included as an appendix for people to review as required.

Comment 11. When were the responders approached to complete the questionnaire? Was it during their pregnancy? If so, at which trimester? We apologise if this was note made clear in the manuscript.

Response 11. This has been clarified in the methods: “Women currently living in Australia and who were currently or had previously experienced severe NVP or HG were eligible to complete the survey. Women could be pregnant at the time of completing the survey.”

We have included a description of the study cohort in the results: “With respect to their most recent pregnancy affected by HG, 110 (38%) indicated they were currently pregnant and experiencing HG, 47 (16%) that their pregnancy ended <6 months ago, 43 (15%) that it ended 6-12 months ago, 39 (14%) that it ended 1-2 years ago and 50 (17%) that it ended >2 years ago.”

Comment 12. The results are presented in a clear and organized manner.

Response 12. No amendments necessary

Comment 13. The discussion is thorough and adequately addresses the key findings of the study.

Response 13. No amendments necessary

Reviewer 2

Comment 14. The authors reported results about an online, national survey of 289 Australian women who were currently or had previously experienced severe NVP or HG, distributed through the HG consumer group, Hyperemesis Australia between July and September 2020. They showed a large variability in antiemetic use and outcomes, highlighting the need for individualized care and treatment approaches during pregnancy.

The subject is interesting because very few published data reported results about national survey in women with NVP or HG. Nevertheless, the study contains many flaws/bias that decrease the interest for the study and major modifications should be done for more clarity.

First, a BIG bias: the survey was realized in women with severe NVP or HG through the HG consumer group, Hyperemesis Australia. Some affirmations were quite evident: “More than 50% of respondents reported major impacts...”, and some affirmations should be written with caution and more discussed: “54% reporting considering terminating their pregnancy and 90% having considered having no more children”. Nevertheless, only 59% of included women were classified as severe NVP using the PUQE score. So, who are the included women????? Are the conclusions adapted???

Response 14. Please note that at the time of conducting the survey there was no agreed definition for severe NVP or HG. It is also important to be aware that terminology such as severe NVP and HG are often used interchangeably, particularly among lay people and patients. While PUQE may be useful as a tool in assisting the identification and management of NVP symptoms, it does not feature as part of the diagnostic criteria included in the Windsor definition for HG so should not be relied on for a diagnostic purpose.

As described in the manuscript, the Windsor definition of HG consists of: start of symptoms in early pregnancy (before 16 weeks gestational age); nausea and vomiting, at least one of which severe; inability to eat and/or drink normally; strongly limits daily living activities. If retrospectively applied, all study respondents would meet these criteria, providing reassurance that this survey indeed was of people with severe NVP or HG.

We don’t agree that restricting our study to women with severe NVP or HG represents a source of bias, as the intent of the study was to evaluate the experiences and perspectives of this cohort regarding medication treatments. In all instances (title, abstract, main text) we have been careful to highlight that findings relate to a cohort of women with severe NVP or HG.

A more important source of bias would be that those who chose to respond to the survey may differ in some way from those that chose not to respond. For example, people dissatisfied with the medication treatments they received may have been more likely to participate in the survey. We have acknowledged this in the study limitations: “Our survey used non-probabilistic sampling and snowballing sampling techniques, making it difficult to extrapolate findings to the broader Australian pregnant population. Those who chose to respond to the survey may differ in some way from those that chose not to respond.”

Comment 15. Second, the recruitment method induced another bias in results such as treatments used in fist line (91% ondansetron and 95% in first trimester) because only severe NVP or HG were included. Many of included women had also experienced HG in a previous pregnancy with ondansetron (64%) as a second or a third line treatment in this previous pregnancy and was then directly used ondansetron as first line in a new pregnancy with HG. It is quite logical!

Response 15. As previously discussed, we have acknowledged the study limitation that results are not necessarily generalisable to all pregnant women irrespective of NVP severity. We have been fully transparent in highlighting that this is a survey of women with severe NVP or HG and make no inferences about how findings relate to non-severe NVP.

In the process of responding to reviewer comments, we identified the opportunity to ensure we provided comparison to previous literature of like studies. We have compared our findings to those of the online survey conducted by Mares et al of women with HG. We have also compared our findings to those of Heitmann et al, who stratified their survey results by NVP severity (mild, moderate, and severe).

Comment 16. The authors should clarify who were the included women: severe NPV or HG, only HG, all women with NVP..). In abstract, in the conclusion, “in cases of severe NVP or HG” should be added to recontextualize. In methods, in “study design”, it is written: “Women currently living in Australia and who were currently or had previously experienced HG were eligible to complete the survey”. Severe NVP as an inclusion criterion has been deleted!

Response 16. We apologise for the typographical error in the methods. This should have read as “Women currently living in Australia and who were currently or had previously experienced severe NVP or HG were eligible to complete the survey” – Please be aware that terminology such as severe NVP and HG are often used interchangeably, particularly among lay people and patients. For that reason we included both terms and this is consistent with the inclusion criteria outlined in the online survey, a copy of which was attached as an appendix to the manuscript.

Comment 17. The discussion is too long, should be shortened, and bias should be more discussed.

Response 17. We have carefully r

---

## [Editor Report · Decision Letter 1]

Assessing the burden of severe nausea and vomiting of pregnancy or hyperemesis gravidarum and the associated use and experiences of medication treatments: an Australian consumer survey

PONE-D-25-16415R1

Dear Dr. Grzeskowiak,

We’re pleased to inform you that your manuscript has been judged scientifically suitable for publication and will be formally accepted for publication once it meets all outstanding technical requirements.

Kind regards,

Shipra Sonkusare, MD, DNB, MNAMS, FICOG, MRCOG, FRCOG

Academic Editor

PLOS ONE